# SciNav: A General Agent Framework for Scientific Coding Tasks

**Tianshu Zhang**
The Ohio State University
zhang.11535@osu.edu

**Huan Sun**
The Ohio State University
sun.397@osu.edu

## Abstract

Autonomous science agents, built on large language models (LLMs), are increasingly being investigated to generate hypotheses, design experiments, and produce reports. Prior science agents primarily focus on open-ended scientific problems, where such outputs—hypotheses, experiments, or analyses are inherently subjective and thus difficult to evaluate rigorously. In contrast, existing scientific coding benchmarks provide tasks with clearly defined, executable outputs that enable objective assessment. However, current agent-based approaches to these benchmarks remain engineering-driven pipelines, lacking structured framework design. This mismatch exposes a gap: the absence of end-to-end, structured science agent frameworks for scientific coding tasks. We address this gap by focusing on scientific coding tasks, where evaluation can be made rigorously, and introducing an agent framework SciNav (Scientific Navigator) that enables more effective solution exploration. Our framework is designed to operate under constrained search budgets, moving beyond reliance on pre-defined success metrics and prolonged search cycles. Inspired by findings that comparative judgments often reveal finer-grained quality differences and therefore provide greater discriminative power than absolute scoring, our framework leverages pairwise relative judgments within a tree search process to select top-K promising solution branches, prune low-potential ones, and progressively narrow down the solution candidates on the selected branches guided by relative comparisons. We demonstrate our agent's effectiveness across different types of tasks on two benchmarks. Experiments show that SciNav significantly outperforms direct prompting and prior agents like OpenHands and Self-Debug across different base models, task types, and difficulty levels, and exceeds different frontier comparators such as random selection and LLM absolute scoring. These results confirm the strength of our agent design and highlight the effectiveness of relative judgment–guided top-K search for high-quality scientific coding, marking a step toward more practical science agents.[1]

## 1 Introduction

Large language models (LLMs) have recently shown strong potential to advance scientific discovery, giving rise to science agents that aim to automate the research process end-to-end. Science agents such as Agent Laboratory (Schmidgall et al., 2025), ResearchAgent (Baek et al., 2024), and AlphaEvolve (Cui et al., 2021), aspire to generate research ideas, design experiments, and draft reports. While this vision of broad-scope automation is compelling, it raises a central challenge: how to evaluate the outputs of such agents. Unlike standard benchmarks with unambiguous correctness criteria, the artifacts produced, including novel hypotheses, experimental protocols, and written analyses, are inherently open-ended and subjective, often demanding expert review or costly human studies to judge their scientific validity.

At the same time, existing scientific coding benchmarks such as DSBench (Jing et al., 2024), DA-Code (Huang et al., 2024), BixBench (Mitchener et al., 2025), DiscoveryBench (Majumder et al.), Core-Bench (Siegel et al., 2024) and SciCode (Tian et al., 2024) define tasks with executable outputs that allow objective assessment. However, current agent-based approaches to these benchmarks

---

[1]Our code is publicly available at https://github.com/OSU-NLP-Group/SciNav

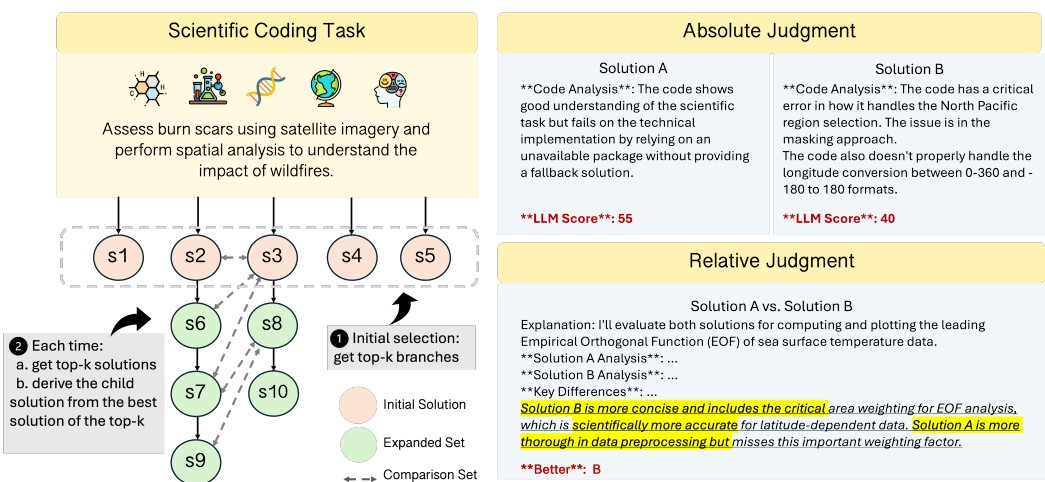

Figure 1: Illustration of Top-K Comparative Tree Search (TKCTS). Left: the search tree expands candidate solutions from an initial set, with relative comparisons (grey dashed arrows) guiding which branches and solutions to retain and explore. Right: comparison between absolute scoring, which LLM assigns pointwise scores to individual solutions, and relative judgment, which LLM evaluates solution pairs and provides sharper, more reliable distinctions. Relative judgments guide the search toward higher-quality solutions under constrained budgets.

largely rely on general-purpose agents such as OpenHands (Wang et al., 2024) and Auto-GPT (Yang et al., 2023), or primarily target engineering-oriented workflows like designing tools for Bash-based environment management and reading or writing files (Huang et al., 2024; Mitchener et al., 2025; Jing et al., 2024). Though these workflows can be adapted to the scientific coding benchmarks, they mainly focus on the engineering pipeline, which lack structured agent framework design that enables more effective solution exploration.

As a result, a gap remains: existing science agents are primarily designed to operate end-to-end on open-ended scientific problems, focusing on generating hypotheses, experimental designs, and reports, while scientific coding benchmarks—with clearly defined, executable outputs that enable direct evaluation, are typically approached using general-purpose agents or engineering-centric pipelines that lack structured framework design. Our work addresses this gap by focusing on scientific coding tasks, where evaluation can be made rigorous, and by introducing a framework that enables more effective solution exploration. Recent efforts such as AIDE (Jiang et al., 2025) have taken steps in this direction by proposing specialized frameworks for machine learning coding tasks. However, these systems often assume (i) well-defined evaluation metrics (e.g., leaderboard accuracy) that agents can directly optimize, and (ii) large exploration budgets (e.g., 24-hour attempts) that allow exhaustive solution generation and testing. Such assumptions are rarely practical, especially when task types are diverse: different scientific problems demand distinct evaluation criteria, many of which are not known in advance, and prolonged search cycles are prohibitively costly. In contrast, our framework produces high-quality solutions under constrained search budgets without relying on pre-specified metrics to provide feedback during the agent exploration.

To address the above limitations, We introduce SciNav (Scientific Navigator), an autonomous agent for scientific coding tasks that leverages Top-K Comparative Tree Search (TKCTS) to effectively navigate solution spaces under constrained budgets. Many studies have shown that relative judgments have higher reliability and discriminative power than absolute scoring (Yan, 2024; Liu et al., 2024; Peyrard et al., 2021). As Figure 1 shows, it's easier to compare two outputs and decide which is better than to assign an absolute score, as comparisons provide clearer signals of where one solution succeeds or fails relative to another. By treating one solution as an anchor against which another is evaluated, comparative assessments sharpen distinctions between candidates, leading to more reliable outcomes and closer alignment with task instruction than pointwise scoring. Building on these findings, TKCTS integrates comparative judgments into a structured tree search. In the early stages, the search explores broadly but retains only the Top-K most promising branches by comparative

| Science Agents | Scope | Multi-Plan? | Retrieval? | Self-Improve Selection | Artifact Evaluated | Automatic Evaluation |
|---|---|---|---|---|---|---|
| ResearchAgent (Baek et al., 2024) | Full-cycle, open-ended | N | Y | NeurIPS review | Idea + Exp. | NeurIPS review |
| AgentLab (Schmidgall et al., 2025) | Full-cycle, open-ended | N | Y | LLM absolute score | Exp. + Paper quality | NeurIPS review |
| CodeScientist (Jansen et al., 2025) | Full-cycle, open-ended | N | N | - | Paper + Code | Accept/Reject Hyp. |
| AI Scientist-V2 (Yamada et al., 2025) | Full-cycle, open-ended | Y | N | Task-defined metric | Paper | NeurIPS review |
| AlphaEvolve (Novikov et al., 2025) | Full-cycle, open-ended | N | N | Task-defined metric | Code Exec. | Task-specific |
| SciMaster (Chai et al., 2025) | QA with tool use | - | Y | - | Text answer | Accuracy |
| AIDE (Jiang et al., 2025) | Machine learning tasks | Y | N | Task-defined metric | Code Exec. | Accuracy, F1 etc |
| SciNav (Ours) | Scientific coding tasks | Y | N | LLM relative judgments | Code Exec. | Task-specific |

Table 1: A comparison of existing science agents with `SciNav`, in terms of their focus, research artifacts being evaluated, and automatic evaluations. 'Full-cycle' refers to the stages from literature review, ideation, experimentation to report writing. 'Self-Improve Selection' refers to the evaluation strategy used to choose solutions from the candidate pool for further refinement. Unlike prior agents that rely on task-defined success metrics, our agent is designed for practical scenarios where such criteria are not available at run time. 'Y' indicates yes; 'N' indicates no; '–' indicates not applicable.

judgments, pruning low-potential ones to control cost. As the search progresses, child nodes are generated along the retained branches, and comparative judgments rank frontier nodes so that the Top-K solutions are promoted for further expansion while the rest are discarded, until a final solution is reached. This design balances systematic exploration with adaptive prioritization, effectively directing computation toward high-quality solutions.

In summary, our contributions are three-fold:

- We introduce `SciNav`, an autonomous science agent for scientific coding tasks that leverages relative judgments within a Top-K tree search to efficiently explore solution spaces under constrained budgets, enabling systematic reasoning and high-quality code generation.

- We conduct a detailed analysis of each component to better understand how it affects the performance of our agent framework.

- Our results demonstrate that `SciNav` achieves substantial improvements over baselines on ScienceAgentBench across different base models. Compared to the strongest baseline agent, Self-Debug, `SciNav` delivers up to a 24% relative gain in success rate (SR) and 7.8 absolute points improvement in VER, and it surpasses OpenHands by 22.9% relative gain in SR. Besides, on DA-Code, `SciNav` achieves 29 absolute points gain on data manipulation and statistical analysis tasks, a 13 absolute points improvement on average and 10%~23% absolute gain across different task difficulties over baselines.

## 2 RELATED WORK

**Science Agents.** To situate our work, we compare existing science agents along their scope, planning mechanisms, retrieval usage, evaluation strategies, and target artifacts. As shown in Table 1, prior systems largely operate in a full-cycle, open-ended setting, aiming to cover the entire research pipeline that produces outputs such as hypotheses, experiments, or reports, which resist systematic automatic evaluation (Baek et al., 2024; Schmidgall et al., 2025; Jansen et al., 2025; Yamada et al., 2025; Cui et al., 2021). Some efforts focus on narrower domains (e.g., machine learning coding tasks) and primarily rely on commonly used task-defined metrics (e.g., accuracy and F1) that assume well-specified success criteria (Jiang et al., 2025), or focus on simple question answering tasks and integrate tool use and retrieval to produce final answers (Chai et al., 2025). In contrast, our work aims to have a principled agent framework to help solving scientific coding tasks, where solutions take the form of executable programs that can be rigorously assessed against ground truth outputs. We present `SciNav`, an agent that integrates relative judgments–guided top-K tree search to navigate solution spaces under constrained computational budgets effectively.

**Test-time Scaling in LLMs.** Prior work such as PlanSearch (Wang et al., 2025) and CodeMonkeys (Ehrlich et al., 2025) demonstrates that increasing the number of generated candidate solutions leads to an approximately log-linear improvement in the proportion of problems successfully solved by at least one candidate. This test-time compute scaling phenomenon significantly enhances overall solution coverage, that can be measured by metrics such as Pass@K in code generation tasks. SFS (Light et al., 2025) reveals performance gains on programming tasks by enhancing the solution diversity and leveraging prior search experiences. Snell et al., 2025 shows that scaling LLM test-time

compute optimally can be more effective than scaling model parameters. Inspired by these work, we leverage the test-time scaling across multiple components of our agent. In addition, we uniquely leverage relative judgments-based frontier comparator to select and refine candidate solutions during tree search. This enables effective exploration and improvement without requiring explicit success criteria, making our approach more generalizable to real-world scientific tasks where such gold signals are often unavailable.

## 3 AGENT FRAMEWORK

---
**Algorithm 1** Top-K Comparative Tree Search (TKCTS)

---
**Input:** Task $T$; initial candidates $S_0$; comparison budget $B$; beam size $K$
**Output:** Final solution $s^\star$
Initialize a priority queue $Q \leftarrow S_0$ ;          // initialization
**while** $Q \neq \emptyset$ **and** $B > 0$ **do**
    $P \leftarrow \text{SELECTPAIRS}(Q)$ ;      // choose candidate pairs to compare
    **foreach** $(s_i, s_j) \in P$ **do**
        $w \leftarrow \text{COMPARE}(T, s_i, s_j)$ ;    // LLM returns which is better and why
        $\text{UPDATERANKING}(Q, s_i, s_j, w)$ ;    // rank candidates
        $B \leftarrow B - 1$
    $S_{\text{keep}} \leftarrow \text{TOPK}(Q, K)$; $S_{\text{drop}} \leftarrow Q \setminus S_{\text{keep}}$; $\text{PRUNE}(S_{\text{drop}})$ ;  // discard low-potential candidates from priority queue
    $E \leftarrow \text{EXPAND}(S_{\text{keep}})$ ;     // generate children / frontier nodes
    $\text{INSERT}(Q, E)$
$s^\star \leftarrow \text{SELECTFINAL}(Q)$ ;      // return the best among remaining
**return** $s^\star$

---

### 3.1 DESIGN MOTIVATION

Scientific discovery is rarely linear: it depends on exploration, revision, and reflection. Yet, most current reasoning agents make shallow, one-shot decisions, lacking the capacity to backtrack, refine, or evaluate solutions strategically. SciNav is designed to overcome these limitations by treating scientific reasoning as a trajectory-driven process, where multiple candidate paths are explored, compared, and refined. Not all paths are equal: some lead to breakthroughs, others to dead ends. SciNav introduces a relative judgments-guided top-K tree search that compares, prunes and expands solution trajectories.

SciNav is inspired by human-like reasoning processes, where initial rough ideas or high-level plans are generated first, followed by iterative debugging and refinement into more rigorous solutions. This approach allows the agent to progressively improve its solutions through self-correction. The agent explores candidate solutions via a tree search, with a top-K selection strategy that both enables controlled backtracking and reduces the cost of subsequent comparative evaluations. Backtracking allows the agent to revisit earlier solutions and reconsider alternative paths when necessary, while relative judgments provide more reliable guidance on which branches to pursue, how to prioritize exploration, and which solutions merit further refinement. Together, these mechanisms enable more deliberate and adaptive problem-solving than prior one-shot or purely engineering pipeline-based agents.

### 3.2 COMPONENTS

As Algorithm 1 shows: TKCTS emphasizes relative judgments as the primary evaluation signal. In each iteration, candidate pairs are compared, rankings are updated, the Top-K branches or candidates are retained, low-potential branches and candidates are pruned, and children of retained branches are expanded. The loop continues under a comparison budget, yielding a final solution selected by pairwise preference. While Algorithm 1 outlines the overall loop of TKCTS, its effectiveness relies on several interacting components. In what follows, we describe four major components—initial planning and code generation, self-debug, self-improve, and the frontier comparator—that together implement the framework.

**Initial Planning and Solution Generation.** Existing work such as PlanSearch (Wang et al., 2025) and CodeMonkeys (Ehrlich et al., 2025) show that as the number of generated solutions increases, the fraction of problems in a dataset that are successfully solved by at least one candidate often grows approximately log-linearly. This test-time compute scaling effect of generating more candidate solutions can significantly improve the overall success solution coverage such as Pass@K in coding tasks. Hence, we follow (Jiang et al., 2025) to generate a rich pool of diverse solution candidates to give more starting points to increase the agents' exploration success. We first prompt LLM to generate multiple high-level plans, and ensure that previously generated plans are visible to the model to avoid repeated plans. Then for each plan, the LLM generates a corresponding program as a candidate solution. This approach unlocks hidden insights that would be missed by deterministic one-pass inference alone.

**Self-Debug for On-the-Fly Error Correction.** Scientific coding tasks require the agent to produce executable code. `SciNav` incorporates a self-debugging mechanism to leverage code interpreter to detect and repair bugs during tree search. This reflective capability enables the agent to revise faulty steps without discarding entire trajectories.

**Iterative Self-Improve through Reflective Reasoning.** Reasoning is not just about fixing mistakes, but about getting better with each step. `SciNav` employs iterative self-refinement by prompting the model to identify a specific refinement point within a selected frontier solution, based on the task description. This process mirrors how humans iteratively refine solutions, progressively improving them toward correctness and completeness.

**Frontier Comparator.** Effective trajectory selection is critical to the success of `SciNav`. At each step of the search, the agent must decide which frontier solutions to expand, refine, or prune. We design the *Frontier Comparator* around comparative judgments, where candidate solutions are directly contrasted against each other rather than scored in isolation. This relative evaluation provides sharper distinctions, greater stability, and closer alignment with human preferences than noisy absolute scores. Concretely, given a pool of candidate solutions, the comparator selects promising branches through iterative pairwise comparisons. The top-K branches are retained for further exploration, while low-potential ones are pruned. As the search deepens, new child solutions are introduced into the pool and evaluated in the same pairwise manner. At each step, we prioritize only the top-K candidates for relative judgments. This strategy enables controllable backtracking by allowing the search to revisit up to K earlier solutions, while also reducing the cost of pairwise relative judgments every time. Crucially, backtracking ensures that if the currently preferred path stagnates or fails, the agent can return to previously lower-ranked candidates and resume exploration from there. This prevents premature commitment to suboptimal solutions and supports adaptive recovery, making the search more resilient. The mechanism also allows the agent to dynamically allocate its limited budget toward the most promising directions. In summary, the Frontier Comparator transforms trajectory selection from one-shot scoring into a comparative, iterative process that systematically navigates solution spaces under constrained resources. After each round of pairwise judgments, a ranking algorithm is applied to the candidates in the priority queue to update their quality ordering (see Appendix F).

## 4 EXPERIMENTAL SETUP

**Agent-level Baselines.** To evaluate the effectiveness of `SciNav`, we compare it against the following three baselines, each representing a different strategy for program generation and reasoning: 1) Direct Prompting: This baseline uses an LLM to generate a solution in a single pass using a task-specific prompt. It reflects the standard zero-shot setting commonly used in prior work. This method serves as a simple and widely adopted baseline for evaluating initial model generation quality. 2) Self-Debug (Chen et al., 2024): In this baseline, the model generates an initial solution and then attempts to improve it via self-debugging based on the Python interpreter execution feedback. While this method allows for limited reflection, it does not incorporate trajectory search, external evaluation, or ranking. It evaluates the isolated effect of a self-correction loop without broader solution exploration. 3) OpenHands (Wang et al., 2024): OpenHands is a general agent that is designed for multiple domains including Web and software engineering tasks. It builds on the ReAct framework (Yao et al., 2023) to generate the next action based on the previous observation. Instead of directly generating the entire program solution at once, OpenHands gradually finishes the solution step by step.

**Frontier Comparator Baselines.** We compare our Frontier Comparator against several alternative selection strategies: (1) *Random Selection*, which randomly picks a candidate solution from the pool; 2) *LLM-Absolute:* The LLM assigns a numerical score to each candidate solution, and the highest-scoring solution is selected for further refinement. 3) *Rubric-Absolute (Chen et al., 2025):* The LLM scores each candidate according to a structured rubric derived from the ground truth program. These serve as baselines to evaluate the advantage of pairwise comparative judgments.

**Datasets.** To evaluate SciNav in realistic scientific coding scenarios, we use two benchmarks: ScienceAgentBench (Chen et al., 2025) and DA-Code (Huang et al., 2024). ScienceAgentBench is a curated benchmark designed to assess agents' capabilities in scientific discovery. This dataset includes a diverse set of tasks that cover the entire workflow such as model development, data analysis, and visualization, spanning from four scientific disciplines: Bioinformatics, Computational Chemistry, Geographical Information Science, and Psychology & Cognitive Neuroscience. All of our experiments are done on their "without expert-provided knowledge" setting. DA-Code encompasses a diverse set of challenging data wrangling and analytics tasks that require advanced data science programming techniques for intricate data processing and answer derivation. To evaluate the effectiveness of SciNav, we randomly sample 100 tasks across different categories such as data insight, data manipulation, data wrangling, and statistical analysis from DA-Code, while also covering different levels of difficulty.

**Experiment Details.** We experiment with GPT-4o (both the 0513 version and the 1120 version) (OpenAI, 2024), Claude-3.7 (sonnet-20250219-v1:0 version) (Anthropic, 2025) and DeepSeek-R1 (DeepSeek-AI, 2025). Since GPT-4o (2024-11-20) version is much cheaper than GPT-4o (2024-05-13) version, we try both versions for the main experiments, but use GPT-4o (2024-11-20) version for the ablation study and frontier comparator part. For all experiments, we use the same hyperparameters. For temperature, we use 0.5 for code generation and 0.5 for debug, analysis and summary, and 0 when leveraging LLMs to compare or judge the solutions. For top-p, we use top 0.95, and perform 0-shot prompting via the APIs. For each baseline, we run three times to get the mean performance. For each frontier comparator, we run the agent twice to get the mean performance. To constrain the budget, we set the initial solution number to 5, maximum debug step to 3 and total exploration step to 10, which includes the self-improvement step if the budget has not been exhausted by self-debug.

**Evaluation Metrics.** We follow previous work (Chen et al., 2025; Huang et al., 2024) to comprehensively evaluate each generated program. For ScienceAgentBench, we use two key metrics. (1) Valid Execution Rate (VER) measures whether a program can execute without errors. (2) Success Rate (SR) assesses whether the output satisfies the specific task goal, such as passing predefined task success criteria, matching expected predictions, or producing a high-quality visualization. These criteria are implemented as task-specific evaluation programs during the benchmark annotation process. Among the reported metrics, SR (Success Rate) is the most important as it directly reflects task success. For DA-Code, we leverage their evaluation suite, which supports multiple tasks through configurable setups, where each task is uniquely identified and defined with its required outputs, metrics, and options. Their tailored scoring methodology assesses agent performance across diverse outputs such as tables, charts, and machine learning predictions, with metrics customized for each output type. (3) API Cost (Cost) reports the average dollar cost required to complete a single task using the agent. This metric accounts for API usage and serves to highlight the importance of designing cost-efficient agents, as emphasized by Kapoor et al., 2024.

## 5 RESULT ANALYSIS

### 5.1 MAIN RESULTS

Table 2 compares the performance of different agent strategies using two versions of GPT-4o (2024-05-13 and 2024-11-20) across two metrics: Success Rate (SR) and Valid Execution Rate (VER). The experiment results show that our proposed agent SciNav consistently outperforms baseline approaches across both model versions. Under the 2024-05-13 model, SciNav achieves an SR of 16.1%, surpassing Self-Debug (14.7%) and OpenHands (13.1%), while maintaining a strong VER of 66.0%. Although the cost (0.512) is higher than Self-Debug (0.057) and Direct Prompting (0.011), SciNav offers a better balance between performance and cost-effectiveness compared to OpenHands (1.093), which is substantially more expensive despite lower SR and VER. With the updated GPT-4o (2024-11-20), due to the constrained budget, we only choose the best baseline Self-Debug for

| Base Model | Agents | SR | VER | Cost ↓ |
|---|---|---|---|---|
| GPT-4o (2024-05-13) | Direct Prompting | 7.50 | 42.2 | 0.011 |
| | OpenHands | 13.1 | 62.8 | 1.093 |
| | Self-Debug | 14.7 | 71.2 | 0.057 |
| | SciNav (Ours) | **16.1** | 66.0 | 0.512 |
| GPT-4o (2024-11-20) | Self-Debug | 15.0 | 67.0 | 0.030 |
| | SciNav (Ours) | **18.6** | **69.9** | 0.342 |
| Claude-3.7 | Self-Debug | 22.5 | 84.3 | 0.066 |
| | SciNav (Ours) | **25.5** | 72.5 | 0.893 |
| DeepSeek-R1 | Self-Debug | 18.6 | 59.8 | 0.023 |
| | SciNav (Ours) | **19.6** | **67.6** | 0.298 |

Table 2: Mean performances of each agent on ScienceAgentBench (Chen et al., 2025). Among the reported metrics, SR (Success Rate) is the most important as it directly reflects task success. VER (Valid Execution Rate) indicates whether a program can be executed without errors and is closely related to the number of debugging steps. Note that we run SciNav with 3 debug steps, while Self-Debug is run with 10 debug steps, which may occasionally allow Self-Debug to achieve higher VER than SciNav.

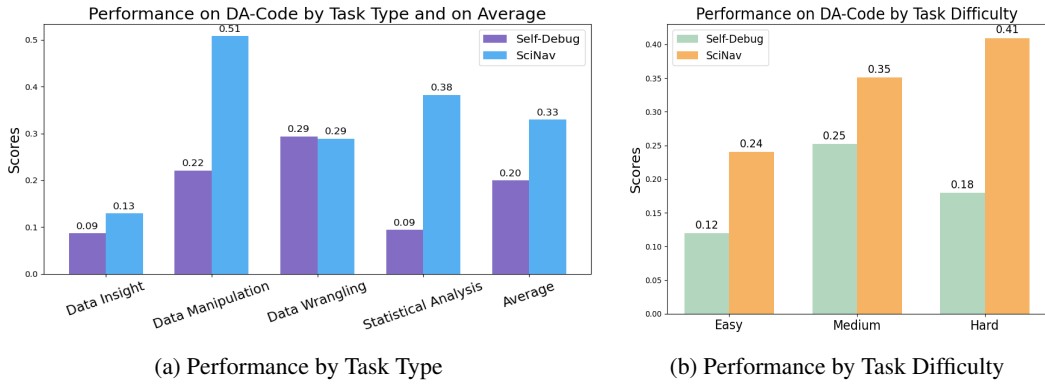

(a) Performance by Task Type          (b) Performance by Task Difficulty

Figure 2: Performance on DA-Code (Huang et al., 2024): (a) by task type, (b) by task difficulty. Model: GPT-4o (2024-11-20).

comparison. SciNav achieves a large performance gain compared to Self-Debug, with a 24% relative improvement for SR and a 2.9-point absolute gain for VER.

**Generalization to Other Base Models.** We further evaluate SciNav on Claude-3.7 and DeepSeek-R1 to assess its generalization beyond GPT-4o. Claude-3.7 is recognized for strong code generation, while DeepSeek-R1 is recognized for strong reasoning. As shown in Table 2, SciNav consistently outperforms Self-Debug on both models, with gains of up to 3 points in success rate (SR) and 8 points in VER. These results indicate that SciNav generalizes effectively across different base models, despite their distinct strengths.

**Generalization to Other Datasets.** Figure 2 presents the performance comparison between SciNav and Self-Debug on DA-Code across different task dimensions. Figure 2a shows that SciNav consistently outperforms Self-Debug in most task categories and on average, achieving notable gains in data manipulation (29% absolute improvement), statistical analysis (29% absolute improvement) and on average (13% absolute improvement), while maintaining comparable performance in data wrangling. Figure 2b further demonstrates that SciNav adapts well to varying task difficulty, yielding strong improvements across easy (12% absolute improvement), medium (10% absolute improvement), and especially hard tasks (23% absolute improvement). These results confirm that SciNav delivers superior performance across diverse task types and task difficulty, underscoring its effectiveness as a scientific coding agent for DA-Code.

| Models | Frontier Comparator | SR | VER |
|---|---|---|---|
| | Random Selection | 15.2 | 64.7 |
| GPT-4o (2024-11-20) | LLM-Absolute | 16.2 | 69.1 |
| | Relative Judgments | **18.6** | **69.9** |
| GPT-4o (2024-11-20) | Rubric-Absolute (w/ GT) | 21.1 | 74.5 |

Table 3: The effect of different frontier comparators on our agent. "w/ GT" means using ground truth program information in the frontier comparator. We include Rubric-Absolute only as a reference, since it is impractical in real-world settings.

## 5.2 FRONTIER COMPARATOR ANALYSIS

Table 3 evaluates the impact of different frontier comparators on agent performance using GPT-4o (2024-11-20). Among the Random Selection, LLM-Absolute and Relative Judgments, Relative Judgments achieves the highest SR (18.6%) and VER (69.9%), outperforming both other two baselines, which attain lower SR of 15.2% and 16.2%, respectively and lower VER of 64.7% and 69.1% respectively.

The Rubric-Absolute achieves the highest SR and VER among all frontier comparators, benefiting from access to ground truth rubric descriptions that detail the correct solution steps. While this method partially leverages ground truth signals, it highlights the potential performance gains from incorporating reliable supervision when available. However, such rubric-based evaluations are often impractical in real-world scientific tasks, where detailed grading criteria are rarely accessible. In contrast, our relative judgments-based frontier comparator offers a more generalizable solution, as it operates independently of ground truth labels while still delivering strong performance. Due to its effectiveness and applicability in reality, we adopt relative judgments as the default frontier comparator in our main agent framework.

## 5.3 COMPONENT ABLATION STUDY

| Num of Init. Solutions | Use Self-Improve? | Avg # of Successful Init. Solutions | Avg # of Successful Nodes | Success Rate |
|---|---|---|---|---|
| 1 | No | 0.24 | 0.40 | 40.5 |
| 2 | No | 0.40 | 0.57 | 40.5 |
| 5 | No | 0.98 | 1.14 | 45.2 |
| 5 | Yes | **1.17** | **2.69** | **57.1** |

Table 4: Component ablation study on `SciNav`. Model: GPT-4o (2024-11-20). The experiment is conducted on 40 tasks of ScienceAgentBench, each of which has been successfully solved at least once by either a baseline agent or `SciNav`.

Table 4 presents a component-wise ablation study evaluating the impact of the number of initial solutions and the use of the self-improvement mechanism in our agent framework. We observe that:

(1) *Number of successful initial solutions directly contributes to end-to-end success rate.* The table shows a strong correlation between the average number of successful initial solutions and the overall success rate. Without self-improvement, increasing the number of initial solutions from 1 to 5 yields a steady improvement in the average number of successful initial solutions (from 0.24 to 0.98), the average number of successful nodes (from 0.40 to 1.14) and the overall success rate (from 40.5% to 45.2%). This indicates that generating multiple initial solutions increases the chance that at least one initial solution is close to correct, thus improving the agent's final performance. The more successful starting points the agent has, the more likely it is to select or build upon a valid reasoning path.

(2) *Self-improvement enables the agent to generate more correct programs and achieve the highest success rate.* The final row of Table 4 isolates the effect of enabling self-improvement: for the same initial solution size 5, when self-improvement is disabled (row 3), the agent achieves an average of 1.14 successful nodes and 45.2% success rate, while when self-improvement is enabled (row 4), the number of successful nodes jumps to 2.69, and the success rate increases significantly to 57.1%. This

demonstrates that self-improvement nearly doubles the number of correct programs, allowing the agent to refine and expand upon flawed or incomplete initial solutions. The improvement in both node-level correctness and overall task success confirms that self-improvement is a key driver of end-to-end performance.

Overall, the findings highlight the complementary roles of initial solutions and iterative self-refinement in enhancing agent performance on the tasks.

## 5.4 ERROR ANALYSIS

To evaluate the effectiveness of our different frontier comparators, we conducted an error analysis on 20 randomly selected unsuccessful tasks from ScienceAgentBench using `SciNav`. We categorize errors into two main types: exploration errors and verification errors. An exploration error occurs when the agent's entire trajectory fails to produce any solution that meets the success criteria. In contrast, a verification error arises when at least one successful solution exists in the trajectory, but the agent fails to identify or select it as the final output.

Figure 3: Error analysis on 20 randomly selected unsuccessful tasks of ScienceAgentBench.

We further subdivide exploration errors into two categories: not executable, where all generated solutions are buggy, and executable but unsuccessful, where some solutions are runnable but do not satisfy the task requirements. As shown in Figure 3, only 15.8% of the failures were due to verification errors, suggesting that our relative judgments-based frontier comparator is generally effective at recognizing correct solutions. The remaining 84.2% of failures were attributed to exploration errors, with 15.8% resulting from non-executable programs and 68.4% from executable but incorrect outputs. This analysis highlights that the dominant source of failure lies in the exploration stage. It suggests that more test-time compute is needed in order to cover successful solution in the trajectory, such as increasing the initial solution size or self-improve more steps.

## 6 INFLUENCE OF K ON SciNav

Figure 4 examines how the top-$K$ solution comparison factor influences SciNav's performance and computational cost. As shown in Figure 4a, increasing $K$ initially improves average task accuracy, with $K = 2$ yielding the highest performance across our preliminary study of 30 randomly sampled DA-Code tasks. Beyond $K = 2$, performance gains do not continue—larger $K$ values either plateau or slightly degrade accuracy. Figure 4b illustrates the corresponding cost trend: the number of required LLM calls grows along with $K$, meaning larger $K$ values incur substantially higher computational overhead. Taken together, these results show that $K = 2$ offers the best trade-off between performance and cost, providing the most effective search behavior under constrained budgets. We therefore adopt $K = 2$ as the default setting for our experiments.

## 7 CONCLUSION

In summary, we proposed `SciNav`, a principled autonomous science agent framework for scientific coding tasks. Unlike prior science agents that mainly operate on open-ended scientific problems

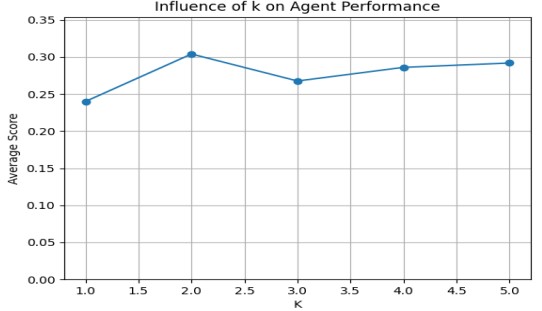 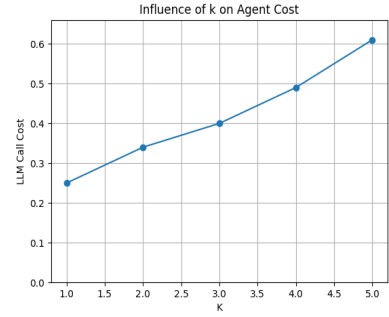

(a) Average score across different values of $K \in \{1, 2, 3, 4, 5\}$ in a preliminary study on 30 randomly sampled DA-Code tasks. $K = 2$ achieves the highest average performance, and is therefore used for all subsequent experiments.

(b) Agent cost across different values of $K \in \{1, 2, 3, 4, 5\}$.

Figure 4: Agent performance and cost across different values of $K \in \{1, 2, 3, 4, 5\}$.

with subjective evaluation criteria, scientific coding tasks enable rigorous assessment through code execution against ground truth. Yet, existing agents for these tasks remain largely general-purpose or depend on purely engineering-driven pipelines. We bridge this gap with a principled agent framework that integrates relative judgments into a Top-K tree search. Our proposed agent `SciNav` systematically explores solution spaces under constrained budgets, enabling both efficient search and high-quality code generation. We provide detailed component-level analysis which shows how each design choice contributes to the overall effectiveness of the framework, offering insights into the mechanisms behind improved agent performance. Through extensive experiments on ScienceAgentBench and DA-Code, we show that `SciNav` consistently outperforms direct prompting and prior agent baselines such as OpenHands and Self-Debug across diverse base models, task types, and difficulty levels, while also surpassing alternative frontier comparators. These results confirm the effectiveness of relative judgment–guided top-K tree search and represent a step toward science agents that are more reliable, principled, and practical.

## ACKNOWLEDGMENTS

The authors would thank colleagues from the OSU NLP group and the Amazon AGI team for constructive feedback. This research was sponsored in part by NSF OAC 2112606, Amazon, Cisco, and Ohio Supercomputer Center (Center, 1987). The views and conclusions contained herein are those of the authors and should not be interpreted as representing the official policies, either expressed or implied, of the U.S. government. The U.S. Government is authorized to reproduce and distribute reprints for Government purposes notwithstanding any copyright notice herein.

## 8 REPRODUCIBILITY STATEMENT

We have made efforts to ensure the reproducibility of our work. The paper provides detailed descriptions of the proposed framework (Section 1), experimental setup, evaluation protocols and implementation details (Section 4) in the main text and Appendix G. All datasets used in our experiments are publicly available, and we describe all the prompts we used for each component in Appendix G. To further facilitate reproducibility, we will release the full source code upon acceptance of the paper.

## 9 ETHICS STATEMENT

This work introduces SciNav, a framework for autonomous science agents tailored to scientific coding tasks. Our study is conducted entirely on publicly available scientific coding benchmarks (ScienceAgentBench and DA-Code) that involve well-defined programming tasks with objectively

verifiable outputs. No human subjects, personal, or sensitive data were used in any part of this research.

We acknowledge the broader ethical considerations of autonomous science agents. While our framework is designed for controlled scientific coding benchmarks, we recognize that autonomous code generation systems may be misused to produce insecure, biased, or harmful code if applied irresponsibly. To mitigate such risks, we restrict our experiments to open benchmarks with safe and bounded tasks, and we clearly document the evaluation setup and intended scope of use.

We affirm that this research complies with the ICLR Code of Ethics. All results and analyses were produced by the authors without external conflicts of interest or undisclosed sponsorship. Our goal is to advance principled, transparent, and rigorously evaluated science agents, while encouraging the community to continue addressing fairness, safety, and responsible deployment in broader applications.

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

## A    LLM Usage Statement

In preparing this manuscript, we used LLM as an assistive tool for polishing the writing. Specifically, the LLM was employed to improve clarity, grammar, and flow of text that had been originally drafted by the authors. The LLM did not contribute to research ideation, experimental design, data analysis, or the development of algorithms. All technical content, results, and conclusions were produced solely by the authors.

## B    Cost Analysis

| Base Model | Agents | SR | VER | Cost ↓ |
|---|---|---|---|---|
| GPT-4o (2024-05-13) | Direct Prompting | 7.50 (0.5) | 42.2 (1.6) | 0.011 (0.000) |
|  | OpenHands | 13.1 (2.6) | 62.8 (2.9) | 1.093 (0.071) |
|  | Self-Debug | 14.7 (3.2) | 71.2 (1.2) | 0.057 (0.001) |
|  | SciNav (Ours) | **16.1** (1.2) | 66.0 (3.5) | 0.512 (0.009) |
| GPT-4o (2024-11-20) | Self-Debug | 15.0 (4.8) | 67.0 (7.4) | 0.030 (0.010) |
|  | SciNav (Ours) | **18.6** (3.3) | **69.9** (0.6) | 0.342 (0.008) |

Table 5: Mean performances of each agent and standard deviations on ScienceAgentBench (Chen et al., 2025). Among the reported metrics, SR (Success Rate) is the most important as it directly reflects task success. VER (Valid Execution Rate) indicates whether a program can be executed without errors and is closely related to the number of debugging steps. Note that we run SciNav with 3 debug steps, while Self-Debug is run with 10 debug steps, which may occasionally allow Self-Debug to achieve higher VER than SciNav.

## C    Statistical Significance Test

We conducted statistical significance test to quantify the reliability of the observed performance differences. Specifically, we compared our agent SciNav with the strongest baseline, Self-Debug, on the DA-Code benchmark using the Mann–Whitney U test. The resulting p-value is 0.0177, indicating that the improvement achieved by our agent is statistically significant ($p < 0.05$).

## D    Cross-Model Evaluation

| Models | | SR | VER |
|---|---|---|---|
| Solution Generation/Refinement | Judge/Verify | | |
| GPT-4o | GPT-4o | 18.6 | 69.9 |
| mixture of GPT-4o and Claude-3.7 | GPT-4o | 19.6 | 63.7 |
| Claude-3.7 | GPT-4o | 18.6 | 72.5 |

Table 6: Model selection of solution generation/refinement and judgment/verification effect on our agent on ScienceAgentBench (Chen et al., 2025).

## E  DA-CODE SAMPLED CASES

We release id of our sampled 100 DA-Code tasks as following: di-csv-005, di-csv-007, di-csv-008, di-csv-009, di-csv-010, di-csv-011, di-csv-012, di-csv-013, di-csv-016, di-csv-018, di-csv-019, di-csv-022, di-csv-023, di-csv-027, di-csv-028, di-csv-029, di-csv-032, di-csv-033, di-csv-034, di-csv-035, dm-csv-002, dm-csv-005, dm-csv-007, dm-csv-014, dm-csv-017, dm-csv-018, dm-csv-019, dm-csv-020, dm-csv-026, dm-csv-029, dm-csv-030, dm-csv-031, dm-csv-035, dm-csv-036, dm-csv-037, dm-csv-039, dm-csv-040, dm-csv-046, dm-csv-054, dm-csv-056, dm-csv-061, dm-csv-062, dm-csv-066, dm-csv-068, dm-csv-069, dm-csv-072, data-wrangling-001, data-wrangling-002, data-wrangling-010, data-wrangling-012, data-wrangling-018, data-wrangling-020, data-wrangling-026, data-wrangling-031, data-wrangling-035, data-wrangling-036, data-wrangling-041, data-wrangling-042, data-wrangling-043, data-wrangling-044, data-wrangling-045, data-wrangling-053, data-wrangling-055, data-wrangling-057, data-wrangling-058, data-wrangling-059, data-wrangling-060, data-wrangling-063, data-wrangling-064, data-wrangling-065, data-wrangling-067, data-wrangling-068, data-wrangling-072, data-wrangling-076, data-wrangling-077, data-wrangling-080, data-wrangling-085, data-wrangling-088, data-wrangling-095, data-wrangling-096, data-wrangling-099, data-wrangling-100, data-wrangling-101, data-sa-001, data-sa-003, data-sa-004, data-sa-009, data-sa-012, data-sa-014, data-sa-016, data-sa-018, data-sa-020, data-sa-022, data-sa-023, data-sa-024, data-sa-030, data-sa-031, data-sa-033, data-sa-039, data-sa-040.

## F  ELO ALGORITHM

For completeness, we provide the details of the Elo Rating algorithm (Elo, 1978) used to maintain a ranking list after relative judgments in SciNav. Firstly, we initialize the Elo score $E_i = 1500$ for each code solution $P_i$. Then, we iteratively update the Elo scores by using the relative scores between any two solutions. Taking the solution pair $P_i$ and $P_j$ as an example, we obtain their comparative judgment results $S_i$ and $S_j$:

**Expected scores:** When solution $P_i$ faces solution $P_j$, the expected score for $P_i$ (denoted $E_i$) is:

$$E_i = \frac{1}{1 + 10^{(R_j - R_i)/400}} \tag{1}$$

Similarly, the expected score for $P_j$ is $E_j = 1 - E_i$.

**After each comparative judgment:**

- If $P_i$ is better: $S_i = 1$, $S_j = 0$
- If $P_j$ is better: $S_i = 0$, $S_j = 1$
- If $P_i$ and $P_j$ are equally better: $S_i = 0.5$, $S_j = 0.5$

**Rating update rule:** Use the standard Elo update formula. For solution $P_i$:

$$R'_i = R_i + K \cdot (S_i - E_i) \tag{2}$$

**Where:**

- $R_i$ is the old rating, $R'_i$ is the new rating.
- $S_i$ is the actual score.
- $E_i$ is the expected score.
- $K$ is a constant. We set it to 32 to determine how fast ratings change.

For each pair of solutions, we update both of their Elo scores once. After all pairwise comparisons, we obtain the final Elo scores for all solutions, which can be used to derive a ranking.

## G  PROMPTS

## Initial Planning and Code Generation

You are a scientific coding expert to write code based on the task description. You need to come up with an excellent, reasonable and creative plan for a solution and then implement this solution in Python. We will now provide a description of the task.

Task description: {}

Task goal: You are an expert Python programming assistant that helps scientist users to write high-quality code to solve their tasks. Given a user request, you are expected to write a complete program that accomplishes the requested task and save any outputs in the correct format. Please wrap your program in a code block that specifies the script type: python. For example:

```python
print("Hello World!")
```

Please keep your response concise and do not use a code block if it's not intended to be executed.
Please do not suggest a few line changes, incomplete program outline, or partial code that requires the user to modify.
Please do not use any interactive Python commands in your program, such as '!pip install numpy', which will cause execution errors.

Memory: {previous generated plans, execution results and summaries}

Response format:
Your response should be a brief outline/sketch of your proposed solution in natural language (3-5 sentences), followed by a single markdown code block (wrapped in ```) which implements this solution and if it's a machine learning task, then prints out the evaluation metric. There should be no additional headings or text in your response. Just natural language text followed by a newline and then the markdown code block.

Solution sketch guideline:
Take the Memory section into consideration when proposing the design, don't propose the same solution.
If it's a machine learning task, then keep the evaluation the same.
The solution sketch should be 3-5 sentences.
If the task is a machine learning task, propose an evaluation metric that is reasonable for this task.
The data is already prepared and available in the './input' directory. There is no need to unzip any files.
For any provided file path, you should replace it with './input' plus the file name and ignore other dictionary or subdirectionary name in the path.

Implementation guideline:
The code should **implement the proposed solution**. If the task is a machine learning task, you should **print the value of the evaluation metric computed on a hold-out validation set**.
The code should be a single-file python program that is self-contained and can be executed as-is.
No parts of the code should be skipped, don't terminate before finishing the script.
Your response should only contain a single code block.
Be aware of the running time of the code, it should complete within 15 minutes.
All the provided input data is stored in './input' directory. **If there is test data provided for this task, please save the test predictions in a 'submission.csv' file in the './working' directory as described in the task description**. This is extremely important since this file is used for grading/evaluation. DO NOT FORGET THE submission.csv file!
You can also use the './working' directory to store any temporary files that your code needs to create.
The evaluation should be based on 5-fold cross-validation but only if that's an appropriate evaluation for the task at hand.

Installed Packages:
Your solution can use any relevant machine learning packages such as: torchvision, scikit-learn, torch, torch-geometric, lightGBM, timm, xgboost, pandas, numpy, bayesian-optimization, statsmodels. Feel free to use any other packages too (all packages are already installed!). For neural networks we suggest using PyTorch rather than TensorFlow.

Data Overview: {}

## Self-Debug Prompt

You are a scientific coding expert to write code based on the task description. Your previous solution had a bug, so based on the information below, you should revise it in order to fix this bug. Your response should be an implementation outline in natural language, followed by a single markdown code block which implements the bugfix/solution.

Task goal: You are an expert Python programming assistant that helps scientist users to write high-quality code to solve their tasks. Given a user request, you are expected to write a complete program that accomplishes the requested task and save any outputs in the correct format. Please wrap your program in a code block that specifies the script type: python. For example:

```python
print("Hello World!")
```

Please keep your response concise and do not use a code block if it's not intended to be executed.
Please do not suggest a few line changes, incomplete program outline, or partial code that requires the user to modify.
Please do not use any interactive Python commands in your program, such as '!pip install numpy', which will cause execution errors.

Here's the user request you need to work on: {task description}

Previous (buggy) implementation: {}
Execution output:

```
Traceback (most recent call last):
File "runfile.py", line 179, in <module>
pair_plot_img = np.frombuffer(pair_plot.fig.canvas.tostring_rgb(), dtype=np.uint8
    )
AttributeError:'FigureCanvasMac' object has no attribute 'tostring_rgb'. Did you
    mean: 'tostring_argb'?
Execution time: 13 seconds seconds (time limit is 15 minutes).
```

Response format:
Your response should be a brief outline/sketch of your proposed solution in natural language (3-5 sentences), followed by a single markdown code block (wrapped in ```) which implements this solution and if it's a machine learning task, then prints out the evaluation metric. There should be no additional headings or text in your response. Just natural language text followed by a newline and then the markdown code block.

Bugfix improvement sketch guideline:
You should write a brief natural language description (3-5 sentences) of how the issue in the previous implementation can be fixed.
The data is already prepared and available in the './input' directory. There is no need to unzip any files. For any provided file path, you should replace it with './input' plus the file name and ignore other dictionary or subdirectionary name in the path.

Implementation guideline:
The code should **implement the proposed solution**. If the task is a machine learning task, you should **print the value of the evaluation metric computed on a hold-out validation set**.
The code should be a single-file python program that is self-contained and can be executed as-is.
No parts of the code should be skipped, don't terminate before finishing the script.
Your response should only contain a single code block.
Be aware of the running time of the code, it should complete within 15 minutes.
All the provided input data is stored in './input' directory.
**If there is test data provided for this task, please save the test predictions in a 'submission.csv' file in the './working' directory as described in the task description**. This is extremely important since this file is used for grading/evaluation. DO NOT FORGET THE submission.csv file!
You can also use the './working' directory to store any temporary files that your code needs to create.
The evaluation should be based on 5-fold cross-validation but only if that's an appropriate evaluation for the task at hand.

Data Overview: {}

## Self-Improvement Prompt

You are a scientific coding expert to write code based on the task description. You are provided with a previously developed solution below and should improve it in order to further increase the performance. For this you should first outline a brief plan in natural language for how the solution can be improved and then implement this improvement in Python based on the provided previous solution.

Task goal: You are an expert Python programming assistant that helps scientist users to write high-quality code to solve their tasks.
Given a user request, you are expected to write a complete program that accomplishes the requested task and save any outputs in the correct format.
Please wrap your program in a code block that specifies the script type: python. For example:

```python
print("Hello World!")
```

Please keep your response concise and do not use a code block if it's not intended to be executed.
Please do not suggest a few line changes, incomplete program outline, or partial code that requires the user to modify.
Please do not use any interactive Python commands in your program, such as '!pip install numpy', which will cause execution errors.

Here is the user request you need to work on: {task description}

Memory: {previous <plan, code, results>}

Response format:
Your response should be a brief outline/sketch of your proposed solution in natural language (3-5 sentences), followed by a single markdown code block (wrapped in ```) which implements this solution and if it's a machine learning task, then prints out the evaluation metric. There should be no additional headings or text in your response. Just natural language text followed by a newline and then the markdown code block.

Solution improvement sketch guideline:
The solution sketch should be a brief natural language description of how the previous solution can be improved.
You should be very specific and should only propose a single actionable improvement.
This improvement should be atomic so that we can experimentally evaluate the effect of the proposed change.
Take the Memory section into consideration when proposing the improvement.
The solution sketch should be 3-5 sentences.
The data is already prepared and available in the './input' directory. There is no need to unzip any files.
For any provided file path, you should replace it with './input' plus the file name and ignore other dictionary or subdirectionary name in the path.

Implementation guideline:
The code should **implement the proposed solution**. If the task is a machine learning task, you should **print the value of the evaluation metric computed on a hold-out validation set**.
The code should be a single-file python program that is self-contained and can be executed as-is.
No parts of the code should be skipped, don't terminate before finishing the script.
Your response should only contain a single code block.
Be aware of the running time of the code, it should complete within 15 minutes.
All the provided input data is stored in './input' directory.
**If there is test data provided for this task, please save the test predictions in a 'submission.csv' file in the './working' directory as described in the task description**. This is extremely important since this file is used for grading/evaluation. DO NOT FORGET THE submission.csv file!
You can also use the './working' directory to store any temporary files that your code needs to create.
The evaluation should be based on 5-fold cross-validation but only if that's an appropriate evaluation for the task at hand.

Previous solution: {program}

## Feedback Prompt

You are a scientific coding expert to write code based on the task description. You have written code to solve this task and now need to evaluate the output of the code execution. You should determine if there were any bugs as well as report the empirical findings.

Task goal: You are an expert Python programming assistant that helps scientist users to write high-quality code to solve their tasks. Given a user request, you are expected to write a complete program that accomplishes the requested task and save any outputs in the correct format. Please wrap your program in a code block that specifies the script type: python. For example:

```python
print("Hello World!")
```

Please keep your response concise and do not use a code block if it is not intended to be executed. Please do not suggest a few line changes, incomplete program outline, or partial code that requires the user to modify.
Please do not use any interactive Python commands in your program, such as '!pip install numpy', which will cause execution errors.

Here is the user request you need to work on: {task description}

Implementation: {program}

Execution output:

```
Analysis complete. Figure saved to pred_results/dkpes_molecular_analysis_pred.png
Execution time: 13 seconds seconds (time limit is 15 minutes).
```

## Relative Judgments Prompt

Please act as an impartial judge and evaluate the quality of the code responses provided by two AI assistants to the programming question displayed below.

You will be given assistant A's answer, and assistant B's answer. Your job is to evaluate which assistant's answer is better. Avoid any position biases and ensure that the order in which the responses were presented does not influence your decision. Do not allow the length of the responses to influence your evaluation. Do not favor certain names of the assistants. Be as objective as possible. After providing a fine-grained analysis of the differences between the two code responses, output your final verdict and score by strictly following this format:
Rating A: [[1-10]]
Rating B: [[1-10]]
Better: [[A or B]]

[Question] {Task description}

[The start of Assistant A's RESPONSE]{response of assistant A}[The end of Assistant A's RESPONSE]

[The start of Assistant B's RESPONSE]{response of assistant B}[The end of Assistant B's RESPONSE]

You need to use the following output format:
***
Explanation: Here is an explanation.
Rating A: [[1-10]]
Rating B: [[1-10]]
Better: [[A or B]]
***

## LLM-Absolute Prompt

Task description: {}
Response summary: {}
The generated code program is as following: {program}

Please generate a score between 0 to 100 to indicate how good and suitable the generated code matches the request.

You can have the explanation, reasoning or analysis, but please explicitly generate the score using the format **Score: **, e.g., **Score: 37** in your response.

## Rubric-Absolute Prompt

Task description: {}
Response summary: {}
The generated code program is as following: {program}
The grading rubric is provided as JSON: {rubric description}

Please generate a score between 0 to 100 based on the given grading rubric (criteria) to indicate how good and suitable the code solution matches the request.

You can have the explanation, reasoning or analysis, but please explicitly generate the score using the format **Score: **, e.g., **Score: 37** in your response.

