# OpenReview forum: "SciNav: A General Agent Framework for Scientific Coding Tasks"
_ICLR.cc/2026/Conference — ICLR 2026 Poster_

### Official Review · Reviewer_2Rev · 2025-10-31

**Soundness:** 2
**Presentation:** 3
**Contribution:** 2
**Rating:** 4
**Confidence:** 3

**Summary:**

This study introduces SciNav, an LLM-based agent framework designed for scientific coding tasks. SciNav employs Top-K Comparative Tree Search to address these tasks under constrained computational budgets. Experimental results demonstrate that SciNav surpasses baseline methods, achieving up to a 24% increase in success rate and a 7.8-point absolute improvement in valid execution rate.

**Strengths:**

-	This paper addresses a clearly defined and underexplored problem: scientific coding tasks, which connect scientific reasoning with executable code generation and allow for objective evaluation.
-	The proposed Top-K Comparative Tree Search introduces a novel use of relative LLM judgments for iterative code refinement. It shows consistent empirical improvements over strong baselines.
-	The paper is well-organized with detailed component analysis, ablation studies, and transparent experimental setup across multiple benchmarks and LLMs.

**Weaknesses:**

-	The primary concern with this study is the gap between the authors’ claims and their actual contributions: the paper frames SciNav as a general “principled agent framework,” but the work focuses narrowly on code-generation heuristics without formal theoretical grounding or demonstration of broader scientific reasoning.
-	The use of the term “principled” in the title and narrative is inappropriate, as the method is not derived from explicit first principles or formal justification. Currently it is a structured heuristic rather than a principled framework in the scientific sense. I highly recommend the authors to use more appropriate terms.
-	The paper lacks quantitative analysis of computational cost and statistical significance of performance gains brought by the proposed method, making claims of efficiency and reliability only partially supported.

**Questions:**

Please see comments above. One additional question: can scientific coding serve as a valid representation of scientific reasoning tasks? How does the proposed system enhance scientific reasoning in open-ended environments?

---

> ### Author Response · Authors · 2025-11-21
> **Response to Reviewer 2Rev**
>
> We appreciate that the reviewer finds our work addresses a clearly defined and underexplored problem, recognizes the novelty of incorporating relative judgments within our framework, and notes that our experiments demonstrate consistent improvements over strong baselines. We are also grateful for the reviewer’s recognition of the paper’s clear organization, thorough analyses, and transparent experimental setup.
>
> ---
> [The primary concern with this study is the gap between the authors’ claims and their actual contributions: the paper frames SciNav as a general “principled agent framework,” but the work focuses narrowly on code-generation heuristics without formal theoretical grounding or demonstration of broader scientific reasoning.] [The use of the term “principled” in the title and narrative is inappropriate, as the method is not derived from explicit first principles or formal justification. Currently it is a structured heuristic rather than a principled framework in the scientific sense. I highly recommend the authors to use more appropriate terms.]
>
> We appreciate the reviewer’s rigorous thoughts regarding the use of the term principled. To make our contribution more precise, which is a structured heuristic framework design based on empirical insights, we have revised the title, abstract and introduction in our paper to avoid the term principled as the reviewer suggests, and now use structured instead.
>
> ---
> [The paper lacks quantitative analysis of computational cost and statistical significance of performance gains brought by the proposed method, making claims of efficiency and reliability only partially supported.]
>
> We thank the reviewer for this valuable comment regarding the cost–performance trade-off and the statistical significance. We have revised **Table 2 in main body** and added **Table 5 in Appendix B** to show the cost and **Appendix C** to show the statistical significance test in our revised paper. We invite the reviewer to refer to these additions in the revised version. Regarding using the statistical significance test to quantify the reliability of the observed performance differences, specifically, we compared our agent with the strongest baseline, Self-Debug, on the DA-Code benchmark using the Mann–Whitney U test. The resulting p-value is 0.0177, indicating that the improvement achieved by our agent is statistically significant (p < 0.05).
>
> ---
> [Can scientific coding serve as a valid representation of scientific reasoning tasks? How does the proposed system enhance scientific reasoning in open-ended environments?]
>
> We appreciate the reviewer’s thoughtful question. Scientific coding tasks represent an important and tractable subset of scientific reasoning tasks, as they allow for rigorous, execution-based evaluation: the agent must generate correct, executable code in response to scientific instructions. However, they do not capture the full spectrum of scientific reasoning. Certain tasks—such as identifying novel mechanisms, proposing unexplored directions, or synthesizing cross-disciplinary insights—cannot be directly solved through code generation.
>
> For open-ended scientific environments, our framework can be applied whenever the task ultimately requires code as the output. For tasks that do not involve code generation, the debugging component of our framework would no longer be necessary, but the remaining components—search, comparative judgment, and refinement—remain applicable and useful to obtain a better solution. Exploring such extensions is an interesting direction for future work.

---

> > ### Comment · Reviewer_2Rev · 2025-11-26
> >
> > Dear authors,
> >
> > Thank you for your response and the extended evaluation on the cost analysis and other aspects raised by other reviewers. Could you please highlight the revised content outside the appendix as well to improve readability?
> >
> > Additionally, although the results are currently presented in the appendix, some of the most important results should be properly integrated into the main body of the paper to provide a comprehensive system evaluation. I encourage the authors to incorporate these revisions with a clearly stated experimental setup.

---

> > > ### Author Response · Authors · 2025-12-03
> > > **New Response to Reviewer 2Rev**
> > >
> > > Thanks for your comments! We have incorporated the cost analysis and the study on the influence of $K$ into the main body of the paper to improve readability and ensure that key results are more accessible. We have also expanded the experimental section with more detailed evaluations and analyses as recommended.

---

### Official Review · Reviewer_9HVs · 2025-10-31

**Soundness:** 2
**Presentation:** 3
**Contribution:** 2
**Rating:** 4
**Confidence:** 4

**Summary:**

The paper proposes SciNav, an agent for scientific coding problems that frames solution search as a Top-K Comparative Tree Search (TKCTS). Instead of relying on absolute LLM scores or task-specific metrics during exploration, SciNav repeatedly performs pairwise (relative) judgments among candidate programs, prunes low-potential branches, and refines the top-K trajectories via self-debug and self-improvement loops. The system is evaluated on ScienceAgentBench and DA-Code, claiming consistent gains over Direct Prompting, Self-Debug, and OpenHands, with ablations suggesting relative judgments beat random or absolute scoring for frontier selection.

**Strengths:**

1. Clear problem focus: Targets scientific coding where outputs are executable and evaluable, avoiding the fuzziness of “end-to-end science agents.”

2. Methodical framing: TKCTS with self-debug/self-improve is a clean, modular agent design; Algorithm 1 is easy to follow and implement.

3. Relative judgments: Sensible use of pairwise comparisons; the frontier-comparator ablation shows consistent benefits over random and absolute scoring.

4. Cross-dataset evaluation: Evidence on ScienceAgentBench and DA-Code, with task-type and difficulty breakdowns and an informative error analysis.

**Weaknesses:**

1. Compute fairness & unclear budgets.
The paper fixes step counts but does not report token, runtime, or dollar budgets across methods. Without normalized compute, it’s unclear if performance gains stem from the algorithm or simply more compute. Reporting tokens / wall-clock / $ per task and re-running under a fixed compute budget is necessary.

2. Small absolute gains and low success-rate regime.
Improvements are modest (e.g., \~2–4% absolute SR gains) and overall SR remains low (\~15–19%), as seen in Table 2. The practical significance is unclear without confidence intervals, per-task breakdowns, or hypothesis testing.

3. LLM-as-judge bias / circularity risk.
If the same model family both generates and judges, relative scoring may reflect stylistic familiarity rather than correctness signal. This risks overestimating benefit of pairwise comparison. Cross-model judging or position-randomization baselines are missing.

4. Baselines not fully representative.
The paper omits competitive selection/reranking baselines such as tournament selection over best-of-N, MCTS with learned value, or verification-guided heuristics. These are relevant comparisons that could challenge the novelty claim.

5. Sparse statistical reporting.
Only 2–3 runs per setting, no reported CIs, no significance tests, and unclear sampling protocol for DA-Code. More rigorous variance reporting is needed, especially given the stochastic nature of LLM benchmarking.

6. Under-leveraged partial execution signals.
The paper emphasizes “no task-specific metric at run time,” but many tasks allow cheap checks (import/compile, partial test subsets, lints). A hybrid relative-judgment + lightweight execution signal could materially improve the agent — and the omission feels like an avoidable limitation rather than a principled choice.

7. Reproducibility gap.
Prompts are included, but code is “will be released upon acceptance.” Given the importance of queueing, frontier selection, and Elo parameters, anonymized code or pseudocode for comparator internals would improve credibility.

**Questions:**

1. Compute parity:
Can you report tokens, wall-clock time, and $ cost per task for each method? Do results hold under strict compute-budget matching?

2. Judge/model decoupling:
Did you evaluate cross-model judging (e.g., Claude evaluates GPT solutions and vice versa)? If not, please include — this is crucial to rule out style bias.

3. Pair selection policy:
How exactly are candidate pairs chosen for comparison? Uniform random? Score-based? Uncertainty-based? Please add an ablation isolating this choice.

4. Comparator stability:
How sensitive performance is to the Elo update parameters and number of comparison calls? A plot of success-rate vs. comparison budget would help.

5. DA-Code sampling clarity:
How were the 100 DA-Code tasks selected and stratified? Please release task IDs and sampling seeds to enable exact replication.

6. Hybrid signal experiment:
Have you tested combining pairwise judgments with cheap verification signals (static checks, partial tests)? This seems easy to add and directly addresses observed failure modes.

7. Baseline strengthening:
Can you add tournament-selection, majority-vote ranking over best-of-N, or MCTS-style search? If not, please justify why these are not relevant or already covered by TKCTS.

---

> ### Author Response · Authors · 2025-11-21
> **Response to Reviewer 9HVs - Part I**
>
> We appreciate the reviewer’s recognition of our clear problem focus, clean methodological framing, effective comparison strategy, comprehensive cross-benchmark evaluation, and informative ablation study.
>
> ---
> [Compute fairness & unclear budgets. The paper fixes step counts but does not report token, runtime, or dollar budgets across methods. Without normalized compute, it’s unclear if performance gains stem from the algorithm or simply more compute. Reporting tokens / wall-clock / $ per task and re-running under a fixed compute budget is necessary.]
>
> We thank the reviewer for this valuable comment regarding the cost–performance trade-off. We have added **Table 5 in Appendix B** to show this. SciNav indeed incurs a higher cost than Self-Debug ($`$0.30` vs `$0.03`). But this additional cost leads to a 24% relative gain in success rate (SR). The key contribution lies not in simply increasing test-time compute, but in **how** it is scaled. In scientific coding tasks, we ultimately require the agent to return a single workable solution from many generated candidates. Because there is no explicit oracle signal indicating which candidate is correct or promising, and which one should be selected to further refine or as the final delivered solution. This makes a naïve increase in sampling alone is insufficient. Therefore, a smart scaling strategy—including structured exploration and more effective selection by relative judgment —is essential to ensure that additional test-time compute actually translates into higher-quality final solutions.
>
> Moreover, it is not meaningful to enforce a fixed compute budget across all baselines, as the computational cost is inherently tied to each agent’s design. For example, direct prompting requires only a single model call; Self-Debug runs until no further errors are detected or a maximum number of steps is reached; and OpenHands often incurs substantial overhead due to its automatic environment-setup and dependency-resolution process. These methods do not operate under an iterative cycle, and their total cost is determined organically by how each agent completes the task. As a result, imposing an artificial, uniform compute budget would distort their intended behaviour and lead to ​​inappropriate comparisons.
>
> ---
> [Small absolute gains and low success-rate regime. Improvements are modest (e.g., 2%-4% absolute SR gains) and overall SR remains low (~15%-19%), as seen in Table 2.]
>
> Though the gains on ScienceAgentBench benchmark is ~2%-3%, our experiments show that the gains on DA-Code benchmark can be much more significant. **Figure 2a** shows that SciNav consistently outperforms Self-Debug in most task categories and on average, achieving notable gains in data manipulation (29% absolute improvement), statistical analysis (29% absolute improvement) and on average (13% absolute improvement). **Figure 2b** further demonstrates that SciNav adapts well to varying task difficulty, yielding strong improvements across easy (12% absolute improvement), medium (10% absolute improvement), and especially hard tasks (23% absolute improvement). These results confirm that SciNav delivers superior performance across diverse task types and task difficulty, underscoring its effectiveness as a scientific coding agent for DA-Code.
>
> ---
> [Judge/model decoupling: Did you evaluate cross-model judging (e.g., Claude evaluates GPT solutions and vice versa)? If not, please include — this is crucial to rule out style bias.]
>
> We appreciate the reviewer’s question regarding the cross-model judging in TKCTS. To assess the consistency of our agent performance, we conducted this evaluation by varying the model that generates initial candidate solutions. As shown in **Table 6 in Appendix D**, our approaches remain stable. For example, using GPT-4o as the judging model, solutions generated by GPT-4o, Claude-3.7, or a mixture of both consistently produce similar SR trends (18.6, 18.6, and 19.6, respectively), which are all better than the best baseline: Self-Debug: 15.0 SR. These results indicate that TKCTS produces stable results without style bias.
>
> ---
> [Sparse statistical reporting. No significance tests.]
>
> We thank the reviewer for this valuable comment regarding the statistical significance. We have added **Appendix C** to show the statistical significance test in our revised paper. We invite the reviewer to refer to these additions in the revised version. Regarding using the statistical significance test to quantify the reliability of the observed performance differences, specifically, we compared our agent with the strongest baseline, Self-Debug, on the DA-Code benchmark using the Mann–Whitney U test. The resulting p-value is 0.0177, indicating that the improvement achieved by our agent is statistically significant (p < 0.05).

---

> ### Author Response · Authors · 2025-11-21
> **Response to Reviewer 9HVs - Part II**
>
> [More rigorous variance reporting is needed, especially given the stochastic nature of LLM benchmarking.]
>
> We’ve added the variance to **Table 5 in Appendix B** of the paper.
>
> ---
> [Hybrid signal experiment: Have you tested combining pairwise judgments with cheap verification signals (static checks, partial tests)?]
>
> ​​We want to emphasize that in our setting, hybrid signals such as static analysis or partial unit tests are unfortunately not applicable. The scientific coding tasks we study involve Python programs that do not provide meaningful static checks, and the tasks—such as visualization, modeling, or numerical analysis—do not come with unit tests or other inexpensive verification signals. The absence of such cheap signals is actually a key challenge in this domain which we want to solve and one of the primary motivations for our framework: when no low-cost correctness indicator is available, comparative judgments become essential for guiding search.
>
> ---
> [Pair selection policy: How exactly are candidate pairs chosen for comparison? Uniform random? Score-based? Uncertainty-based? Please add an ablation isolating this choice.]
>
> At each refinement step, we form the K comparison set by taking the top-(K-1) candidates from the previous round (the current “winners”) and plus the newly generated child solution. These candidates constitute the pool from which all pairwise comparisons are performed (a ranking sequence derived by the Elo algorithm stated in Appendix G). Regarding selection policies, we have already evaluated several alternatives—uniform random selection, LLM absolute scoring, and rubric-based scoring—as shown in **Table 3**, which provides an ablation isolating the effect of different comparison strategies.
>
> ---
> [Comparator stability: How sensitive performance is to the Elo update parameters and number of comparison calls? A plot of success-rate vs. comparison budget would help.]
>
> We have added **Figure 4a in Section 6** to show the impact of varying K on our agent’s performance. In our framework, K is the beam width in the top-K search: at each expansion step, the agent retains the K most promising candidate solutions for comparison. To study their effect, we vary K in {1, 2, 3, 4, 5}. For each value of K, we allocate the comparison budget required to compare all K candidates at each step, while fixing the maximum search depth to 10. **Figure 4a in Section 6** shows that K=2 leads to the best performance. **Figure 4b in Section 6** shows the comparison budget needed.
>
> ---
> [DA-Code sampling clarity: How were the 100 DA-Code tasks selected and stratified? ]
>
> We’ve added the sampled DA-Code tasks’ ID in the revised paper. Please check **Appendix E**.
>
> ---
> [Baseline strengthening: Can you add tournament-selection, majority-vote ranking over best-of-N, or MCTS-style search? If not, please justify why these are not relevant or already covered by TKCTS.]
>
> We would like to emphasize that comparative judgment is only one component of our method; our primary contribution is the overall agent framework. Accordingly, our main baselines are existing agent systems rather than isolated ranking strategies. Nonetheless, we provide a detailed analysis of the ranking strategies mentioned by the reviewer, as presented below.
>
> Tournament selection is already inherent in TKCTS: candidate solutions on each branch compete through pairwise relative judgments, and only the top-K “winners” advance. Subsequent comparisons operate on the top-K candidates from the previous step rather than on randomly sampled individuals. If we view the top-K set as the selected population, TKCTS effectively embodies the same spirit as tournament selection. Since our primary baselines are existing agent systems, adding a separate tournament-selection variant would be redundant and not closely aligned with the core contribution of our work.
>
> Majority-vote ranking over best-of-N relies on aggregated absolute scoring. As we show in **Table 3**, absolute scoring is less discriminative and aggregating such scores needs multiple attempts, which is more costly under constrained compute budgets. TKCTS already provides a more efficient and finer-grained ranking through iterative pairwise comparisons.
>
> MCTS-style search is conceptually related, and TKCTS can be viewed as a cost-constrained adaptation of MCTS. Classical MCTS requires numeric value estimates and many rollouts, which are very costly due to these rollouts simulation. TKCTS replaces rollout-based evaluation with cheaper relative judgments while preserving the core expand–evaluate–prune cycle.
>
> For these reasons, given our main contribution is the entire agent framework instead of just a comparison module, we believe SciNav already incorporates the key search components the reviewer refers to, and adding full tournament selection, MCTS or majority-vote baselines would be redundant and cost-inefficient.

---

### Official Review · Reviewer_Eeg5 · 2025-10-31

**Soundness:** 3
**Presentation:** 2
**Contribution:** 2
**Rating:** 6
**Confidence:** 3

**Summary:**

The authors propose and evaluate SciNav, a "framework" (agent) that performs scientific coding. The agent performs search taking into account a constrained budget, and uses pairwise (comparative) jugements rather than absolute judgements to guide the search. They show SciNav outperforms prior agents like OpenHands.

**Strengths:**

- The empirical result of outperforming OpenHands and Self-Debug is quite compelling
 - nice to see search budgets taken into account in the framework
 - nice to see you leveraging existing benchmarks rather than creating a new one
 - ablation that suggests relative judgements are helping (a little bit)

**Weaknesses:**

- Gains are somewhat modest (~2%-3%), so the impact of the work seems a little limited
 - Comparison with genetic algorithm approaches to coding (e.g., in AI Scientist) would be useful

Minor:
 - Abstract takes way to long too get to the goal and contribution - should be stated in first or second sentence. (The abstract gives the impression at first you're going to propose a benchmark)
 - Would be worth expanding on use of relative judgements in AI, e.g., it's the basis of A/B testing, preference optimization (e.g., DPO), and other methods.

**Questions:**

- A common approach in other agent-based coding tasks is to use genetic algorithms to merge different coding ideas (e.g., AIScientist), rather than expanding a single parent. It'd be nice to know how your "single parent" approach would compare. Do you have any intuitions about this?
 - It seems that your contribution is more proposing and evaluating TKCTS as a great framework for agent coding, rather than the (narrower) use of comparative judgements. Is that a reasonable reframing of the contribution?

---

> ### Author Response · Authors · 2025-11-21
> **Response to Reviewer Eeg5**
>
> We appreciate that the reviewer finds our experimental results compelling, the ablation study informative, and our consideration of search budgets valuable.
>
> ---
> [Gains are somewhat modest (~2%-3%), so the impact of the work seems a little limited]
>
> Though the gains on ScienceAgentBench is ~2%-3%, our experiments show that the gains on DA-Code can be much more significant. **Figure 2a** shows that SciNav consistently outperforms Self-Debug in most task categories and on average, achieving notable gains in data manipulation (29% absolute improvement), statistical analysis (29% absolute improvement) and on average (13% absolute improvement). **Figure 2b** further demonstrates that SciNav adapts well to varying task difficulty, yielding strong improvements across easy (12% absolute improvement), medium (10% absolute improvement), and especially hard tasks (23% absolute improvement). These results confirm that SciNav delivers superior performance across diverse task types and task difficulty, underscoring its effectiveness as a scientific coding agent for DA-Code.
>
> ---
> [A common approach in other agent-based coding tasks is to use genetic algorithms to merge different coding ideas (e.g., AIScientist), rather than expanding a single parent. It'd be nice to know how your "single parent" approach would compare. Do you have any intuitions about this?]
>
> We appreciate the reviewer’s thoughtful comparison. Genetic or population-based methods (e.g., AIScientist) promote exploration by combining multiple candidate program ideas. Although we did not directly apply AIScientist to our tasks, we experimented with related approaches, including integrating PlanSearch [1], which also attempts to fuse diverse candidate ideas in a “brainstorming”-style manner. In our preliminary experiments, however, these population-style combinations did not outperform our current design. We hypothesize that such fusion can introduce more creative but also more error-prone or hallucinated behaviors, especially under constrained search budgets, leading to less reliable progress than our more focused strategy.
> Our framework instead emphasizes maintaining a small number of diverse candidates while adopting a single-parent refinement strategy, focusing on efficient exploitation — repeatedly improving the most promising branch candidates through comparative judgment. This avoids the combinatorial explosion often associated with genetic approaches, reduces computational cost, and still captures diversity through stochastic generation at each step. We view population-based evolution as a complementary direction, and exploring hybrid designs (e.g., occasional crossovers between multiple strong candidates) is an interesting avenue for future work.
>
> ---
> [It seems that your contribution is more proposing and evaluating TKCTS as a great framework for agent coding, rather than the (narrower) use of comparative judgements. Is that a reasonable reframing of the contribution?]
>
> We thank the reviewer for this observation. Indeed, our main goal is to demonstrate TKCTS as a flexible framework for building a scientific agent coding system, within which comparative judgment plays a central but not exclusive role. The comparative module is an important component, but the broader framework establishes a foundation that enables structured agent exploration and verification behavior.
>
> ---
> [1] Wang, et al. Planning In Natural Language Improves LLM Search For Code Generation. ICLR 2025

---

> > ### Comment · Reviewer_Eeg5 · 2025-11-25
> >
> > Ah thank you for the clarification, I see you do get greater gains on DA-Code, very nice!

---

### Official Review · Reviewer_2AeN · 2025-10-31

**Soundness:** 3
**Presentation:** 3
**Contribution:** 3
**Rating:** 6
**Confidence:** 3

**Summary:**

In this paper, the authors focus on improving scientific coding agents by proposing SciNav, a framework that treats problem-solving as a structured search guided by relative evaluation. They introduce Top-K Comparative Tree Search (TKCTS), which allows the agent to explore, compare, and refine code solutions through pairwise relative judgments rather than absolute scoring. SciNav integrates components for planning, self-debugging, self-improvement, and frontier selection using an Elo-based ranking mechanism. The authors evaluate SciNav on scientific coding benchmarks to demonstrate that this principled, comparison-driven approach leads to more effective and reliable solution generation than the baseline agents.

**Strengths:**

(1) The paper is well-motivated and clearly defines the need for principled frameworks for scientific coding tasks with verifiable outputs.

(2) It presents a structured search method combining relative judgments and iterative refinement, supported by consistent quantitative improvements over existing agent baselines.

**Weaknesses:**

* Evaluation is limited to two controlled benchmarks, leaving uncertainty about generalization to real-world or open-ended scientific tasks.
* Reliance on LLM-as-judge comparisons may introduce bias, as the same models both generate and evaluate solutions.
* The fixed and narrow search budget restricts exploration, and scalability to more complex tasks remains unclear.

**Questions:**

* Was a cost or runtime comparison performed to quantify the additional computation introduced by pairwise judgments and iterative search relative to baselines?
* The TKCTS relies on relative judgments by an LLM-as-judge. How consistent are these judgments across multiple runs or judging models? Would cross-model evaluation (e.g., using a different LLM as the judge) yield stable rankings?
* The framework uses a fixed budget (five initial solutions, three debug steps, ten total exploration steps). Why were these values chosen, and have the authors tested sensitivity to these parameters?
* Given that relative judgments guide the search process, was any validation (e.g., human evaluation or ground-truth correctness checks) performed to verify that the LLM-judge’s preferences align with actual code quality?

---

> ### Author Response · Authors · 2025-11-21
> **Response to Reviewer 2AeN**
>
> We appreciate that the reviewer finds our paper is well-motivated and our experiments show consistent quantitative improvements over existing agent baselines.
>
> ---
> [Evaluation is limited to two controlled benchmarks, leaving uncertainty about generalization to real-world or open-ended scientific tasks.]
>
> We appreciate the reviewer’s concern regarding the scope of our evaluation. First, we emphasize that the tasks used in SAB and DA-Code are realistic and representative: both benchmarks are constructed from real-world scientific repositories and datasets, reflecting genuine workflows, coding patterns, and failure modes encountered in practical scientific programming. This grounds our evaluation in authentic use cases rather than synthetic toy problems.
>
> Second, we intentionally scope our work to scientific coding tasks where rigorous, execution-based evaluation is feasible. This allows us to measure correctness objectively and assess the effectiveness of our methods in a controlled yet realistic setting. Open-ended scientific tasks inherently require domain experts to judge the scientific validity of the agent’s outputs. Conducting such expert-based evaluations at scale is costly and subjective, which is why we restrict our study to scientific coding tasks with executable, objectively measurable outcomes.
>
> ---
> [Reliance on LLM-as-judge comparisons may introduce bias, as the same models both generate and evaluate solutions.]
>
> Our choice of an LLM-as-judge framework is motivated by the need to scale beyond human annotation while maintaining generalization across diverse scientific coding tasks. To mitigate model-specific bias, we’ve also added a cross-model judgment evaluation in which the models of generating candidate solutions and performing judgment are different. Please check **Table 6 in Appendix D** in the revised draft. We also give a detailed description in the following response.
>
> ---
> [The TKCTS relies on relative judgments by an LLM-as-judge. How consistent are these judgments across multiple runs or judging models? Would cross-model evaluation (e.g., using a different LLM as the judge) yield stable rankings?]
>
> We appreciate the reviewer’s question regarding the robustness of LLM-based relative judgments in TKCTS. To assess the consistency of our agent performance, we conducted cross-model evaluation by varying the model that generated initial candidate solutions. As shown in **Table 6 in Appendix D**, our approaches remain stable. For example, using GPT-4o as the judging model, solutions generated by GPT-4o, Claude-3.7, or a mixture of both consistently produce similar SR trends (18.6, 18.6, and 19.6, respectively), which are all better than the best baseline: Self-Debug: 15.0 SR. These results indicate that TKCTS produces stable results regardless of which high-quality LLM serves as the generation model.
>
> ---
> [Was a cost or runtime comparison performed to quantify the additional computation introduced by pairwise judgments and iterative search relative to baselines?]
>
> Yes, we have added the cost analysis. Please check **Table 5 in Appendix B**.
>
> ---
> [The framework uses a fixed budget (five initial solutions, three debug steps, ten total exploration steps). Why were these values chosen?]
>
> We thank the reviewer for highlighting the importance of hyperparameter sensitivity. We conducted preliminary studies to select the most effective settings for SciNav’s exploration budget. Specifically, we evaluated different numbers of initial solutions (1, 2, 5, 10), maximum debug iterations (3 vs. 10), and overall exploration limits (7, 10, and 15 total steps). These experiments showed that five initial solutions, three debug steps, and ten total exploration steps provide a good balance between performance and computational cost. Increasing the budget beyond these values yielded only marginal improvements while increasing the cost, whereas smaller budgets degraded solution quality. Based on these findings, we selected the reported hyperparameters as a practical and effective configuration.
>
> ---
> [Given that relative judgments guide the search process, was any validation (e.g., human evaluation or ground-truth correctness checks) performed to verify that the LLM-judge’s preferences align with actual code quality?]
>
> We appreciate the reviewer’s question. In our setting, the ultimate evaluation of solution quality is based on ground-truth execution correctness, which serves as an objective measure. During development, we verified that in the cases where relative judgments select a higher-ranked candidate, that candidate is more likely to pass the execution tests compared to alternatives.

---

### Official Review · Reviewer_2cow · 2025-10-31

**Soundness:** 2
**Presentation:** 3
**Contribution:** 2
**Rating:** 4
**Confidence:** 3

**Summary:**

This paper introduces SciNav (Scientific Navigator), a framework for autonomous science agents designed to tackle scientific coding tasks. The core contribution is the Top-K Comparative Tree Search (TKCTS) algorithm, which replaces absolute scoring with pairwise relative judgments during solution exploration. SciNav integrates several components: initial multi-plan generation, self-debug, iterative self-improvement, and a frontier comparator based on relative LLM judgments, to progressively refine code solutions under constrained computational budgets.

Experiments on ScienceAgentBench and DA-Code show that SciNav performs best compared to baselines such as OpenHands and Self-Debug. Ablations also show that relative comparison helps compared to random selection and absolute scoring.

**Strengths:**

S1. The relative judgment–guided Top-K search is a well-motivated methodological idea that builds on prior insights about the reliability of pairwise evaluation and applied in an agentic setting.

S2. The experiments are reasonable, covering two benchmarks, several LLM backbones, and detailed component ablations. The experiments for the contributions of each component, including initial plan diversity, self-improvement, and the comparator strategy are appreciated.

**Weaknesses:**

While the results and experiments are good, my main concerns center around how much we can interpret from them which I'm happy to change with some clarification.

First, the paper does not report error bars or statistical significance. This makes it hard to assess whether observed performance differences are meaningful or consistent across runs.

Second, it is important, especially when we consider deployment to also compare the cost of each agent/ablation involved. How many extra LLM calls/tokens are used for SciNav vs. Self-debug to obtain the performance increases? What is the cost of inference time or $ cost to have extra LLM calls?

Without this information, it is hard to assess the tradeoff/value of SciNav. For instance, how much of this performance gain is just a result of scaling test-time compute.

**Questions:**

What is the K in top-K and the comparison budget for the experiments?
How does changing this impact the performance? Given the main contribution is the comparator it would be helpful to see how much performance is impacted by these hyperparamter choices.

---

> ### Author Response · Authors · 2025-11-21
> **Response to Reviewer 2cow**
>
> We are grateful that the reviewer finds our methodology is well-motivated and our experiments are comprehensive.
>
> ---
> [First, the paper does not report error bars or statistical significance.]
>
> We thank the reviewer for this valuable comment. We have added statistical significance test to quantify the reliability of the observed performance differences. Specifically, we compared our agent with the strongest baseline, Self-Debug, on the DA-Code benchmark using the Mann–Whitney U test. The resulting p-value is 0.0177, indicating that the improvement achieved by our agent is statistically significant (p < 0.05). We revised our draft to include error bars representing the variance across multiple runs in the **Table 5 of Appendix B** and include the statistical significance test in **Appendix C**.
>
> ---
> [Second, it is important, especially when we consider deployment to also compare the cost of each agent/ablation involved.]
>
> We thank the reviewer for this valuable comment regarding the cost–performance trade-off. We have added **Table 5 in Appendix B** and revise **Table 2 in main body** to show this. SciNav indeed incurs higher cost than Self-Debug (`$0.30` vs `$0.03`). But this additional cost leads to a 24% relative gain in success rate (SR). The key contribution lies not in simply increasing test-time compute, but in **how** it is scaled. In scientific coding tasks, we ultimately require the agent to return a single workable solution from many generated candidates. Because there is no explicit oracle signal indicating which candidate is correct or promising, and which one should be selected to further refine or as the final delivered solution. This makes a naïve increase in sampling alone is insufficient. Therefore, a smart scaling strategy—including structured exploration and more effective selection by relative judgment —is essential to ensure that additional test-time compute actually translates into higher-quality final solutions.
>
> ---
> [What is the K in top-K and the comparison budget for the experiments? How does changing this impact the performance? Given the main contribution is the comparator it would be helpful to see how much performance is impacted by these hyperparamter choices.]
>
> We thank the reviewer for the question. We have added **Figure 4a in Section 6** to show the impact of varying K on our agent’s performance. In our framework, K is the beam width in the top-K search: at each expansion step, the agent retains the K most promising candidate solutions. To study their effect, we perform a preliminary analysis on 30 randomly sampled DA-Code tasks and vary K in {1, 2, 3, 4, 5}. For each value of K, we allocate the comparison budget required to compare all K candidates at each step, while fixing the maximum search depth to 10. For instance, when K = 3, each time the agent generates a new candidate solution, it performs comparisons among the three active candidates to select the most promising one for further refinement. **Figure 4b in Section 6** shows the comparison budget needed.
>
> The results in Figure 4 show that K=2 achieves the best performance. Also smaller K leads to smaller comparison budgets. Based on this analysis, we select K=2 for all main experiments.

---

### Author Response · Authors · 2025-12-03
**Update Notes for Paper Draft according to Reviewers' Suggestions**

We thank the reviewers for their important suggestions! We've updated our paper include:

* Revise Table 2 in the main body and add Table 5 in Appendix B to show the detailed cost analysis and performance variance. (**2cow, 2AeN, 9HVs**)
* Add Section 6 in the main body for the study on the influence of K on both performance and cost of SciNav. (**2cow, 9HVs**)
* Add Appendix C for the statistical significance test. (**2cow, 9HVs, 2Rev**)
* Add Appendix D for cross-modal evaluation, where the generation model and judgment model are different. (**2AeN, 9HVs**)
* Add Appendix E to show the sampled DA-Code tasks’ ID. (**9HVs**)
* Integrate important appendix contents into the main body and add more detailed experiment evaluation and analysis. (**2Rev**)

We believe these updates help address the reviewers’ main concerns regarding the experiments.

---

### Meta-Review · Area_Chair_GqmA · 2026-01-07

**Summary:**

Reviewers broadly agree that SciNav targets an important and well-defined problem of scientific coding with executable correctness, where Top-K Comparative Tree Search (TKCTS) is a sensible and well-engineered way to guide agent search using relative judgments instead of unreliable absolute scores. The main points of contention center on evaluation rigor and interpretation rather than correctness: cost and compute fairness, statistical significance, judge bias, baseline completeness, and whether the gains justify the added complexity. The rebuttal is comprehensive and responsive, adding statistical tests, variance reporting, cost analysis, hyperparameter sensitivity studies, cross-model judging, clearer framing of contributions, and revised wording to avoid overstated claims. After rebuttal, most concerns shift from missing evidence to questions of scope, framing, and whether the empirical improvements are impactful enough for broad acceptance.

**Reviewer Concerns:**

Reviewer 2cow: The rebuttal fully addresses concerns about statistical significance, cost–performance tradeoffs, and sensitivity to the Top-K comparison budget by adding significance tests, cost tables, and detailed analyses of how varying K affects performance.

Reviewer 2AeN: The rebuttal addresses nearly all concerns by adding cost analysis, cross-model judging for robustness, hyperparameter sensitivity studies, and a clear justification for benchmark scope, leaving only inherent limitations about generalization to open-ended scientific reasoning.

Reviewer Eeg5: The rebuttal successfully clarifies that gains are substantially larger on DA-Code than initially perceived, provides intuition for comparisons with genetic-style methods, and reframes the contribution as a broader agent framework, fully resolving the reviewer’s questions, which is confirmed by the reviewer’s follow-up comment.

Reviewer 9HVs: The rebuttal addresses most major criticisms by adding cost reporting, significance testing, variance analysis, cross-model judging, hyperparameter sensitivity plots, clearer pair-selection mechanics, and task sampling details, but some skepticism remains about compute-normalized comparisons and the exclusion of certain strengthened baselines, which are argued away rather than empirically closed.

Reviewer 2Rev: The rebuttal resolves the core concerns by revising overstated “principled” claims, adding cost and significance analysis, clarifying the scope of scientific coding as a subset of scientific reasoning, and integrating key appendix results into the main text after follow-up.

**Reviewer Scores:**

Reviewer 2cow did not explicitly state a score change, but since all their concrete methodological and evaluation concerns were directly addressed, their concerns appear fully resolved by the rebuttal.

Reviewer 2AeN did not state a score change, but given that all listed weaknesses were answered with new analyses and clarifications, their concerns are largely resolved.

Reviewer Eeg5 explicitly reacted positively after the rebuttal and indicated improved perception, suggesting their score would likely remain positive or increase.

Reviewer 9HVs did not indicate a score update, and although many concerns were addressed, some reservations about compute fairness and baseline coverage remain partially unresolved.

Reviewer 2Rev explicitly followed up after the rebuttal requesting clearer integration of new results, and after the authors complied, their remaining concerns appear largely resolved.

---

### Decision · Program_Chairs · 2026-01-26

Accept (Poster)